# Impact of Different Storage Temperature on the Enzymatic Activity of *Apis mellifera* Royal Jelly

**DOI:** 10.3390/foods11203165

**Published:** 2022-10-11

**Authors:** Simona Sagona, Francesca Coppola, Gino Giannaccini, Laura Betti, Lionella Palego, Elena Tafi, Lucia Casini, Lucia Piana, Raffaele Dall’Olio, Antonio Felicioli

**Affiliations:** 1Department of Veterinary Sciences, University of Pisa, 56124 Pisa, Italy; 2Department of Pharmacy, Pisa University, 56126 Pisa, Italy; 3Department of Clinical and Experimental Medicine, Pisa University, 56126 Pisa, Italy; 4Consiglio per la Ricerca in Agricoltura e L’analisi Dell’economia Agraria—Centro di ricerca Agricoltura e Ambiente, 40128 Bologna, Italy; 5Piana Ricerca e Consulenza s.r.l. a Socio Unico, 40024 Bologna, Italy; 6BeeSources, 40132 Bologna, Italy

**Keywords:** royal jelly, enzymatic activity, freshness markers, storage, antioxidant enzymes

## Abstract

Royal Jelly is a nutrient secretion of nurse bees and a high interest functional food in human nutrition. Very little information is available on its chemical composition integrity and enzymatic activity during shelf life and assessment of new freshness markers are desirable for its conservation. In this study, the activity of glucose oxidase, five proteases and two antioxidant enzymes in refrigerated and frozen Royal Jelly for different storage times was preliminary investigated. Refrigeration determined a significantly reduction in glucose oxidase and carboxypeptidase A-like activity in Royal Jelly after one year of storage while no differences were recorded in the activity of these enzymes in frozen samples. After one year of storage glucose oxidase and carboxypeptidase A-like activity resulted higher in frozen samples frozen than in refrigerate ones. Results obtained suggest that the activities of these enzymes may be good markers of Royal Jelly freshness within 1 year at refrigeration condition. Freezing could be a valid alternative storage method to ensure a higher preservation of glucose oxidase and carboxypeptidase A-like activities for at least 1 year. Further investigation to determine the timing of glucose oxidase inactivation/degradation under refrigerated conditions and the enzymatic activity trend under prolonged frozen conditions are desirable.

## 1. Introduction

Royal Jelly (RJ) is a nutrient rich secretion, yellowish-white in color, acidic in nature, with a pungent odor and taste, with a gelatinous consistency, and usually not homogenous due to the presence of undissolved particles of variable size [1]. It is secreted by nurse honeybees from their hypopharyngeal, mandibular, post-cerebral and thoracic glands after partial consumption of pollen and nectar/honey [2]. Worker fate larvae fed RJ for only 3 days, and later they fed on a mixture of honey, RJ, and pollen, while in queen bees RJ is the primary food throughout their lifetime [3].

RJ is composed of 60–70% moisture, 9–18% proteins, 7–18% carbohydrates and 3–8% lipids [4]. Aspartic acid is the main amino acid and 10-hydroxy-2-decenoic acid the major fatty-acid [5]. The lipid fraction of RJ consists of 80–85% fatty acids, 4–10% phenols, 5–6% waxes, 3–4% steroids, and 0.4–0.8% phospholipids [6]. Glucose and fructose account for over 90% of the total sugar content in RJ [6]. Recently, thanks to proteomics techniques, RJ proteins were detected, identified and characterized [2,7,8,9,10,11,12,13]. The most soluble RJ proteins are Major Royal Jelly Proteins (MRJPs), of which there are nine members: MRJP1, MRJP2, MRJP3, MRJP4, MRJP5, MRJP6, MRJP7, MRJP8 and MRJP9 [6]. Among them, MRJP1 is the most abundant one [7].

Royal Jelly possesses several health-promoting properties such as antibacterial, anti-inflammatory, vasodilative, hypotensive, disinfectant, antioxidant, antihypercholesterolemic and antitumor activity [14]. Based on its unique composition, there is a growing interest in consumers and food industry towards this functional food, thus far mostly used in dietetics and cosmetics in several countries. Due to the importance of RJ in human nutrition its shelf-life has been widely investigated [8,12,15,16,17]. Furosine, 5-hydroxymethyl-2-furaldehyde, 57-kDa protein, major RJ protein 4, 5 (MRJP4, MRJP5), MRJP1, glucose oxidase, peroxiredoxin and glutathione S transferase were suggested as potential markers for RJ freshness since their abundance could change according to storage condition, time and temperature [2,8,9,11,12,13,14,15,16,17,18]. However, using a proteomic approach, MRJP4, 5, glucose oxidase, peroxiredoxin, and glutathione S-transferase S1 were reported to be absent in RJ after 1 year of storage at room temperature [12]. Glucose oxidase is an enzyme secreted by bee hypopharyngeal glands and involved in the transformation of D-glucose into gluconic acid and hydrogen peroxide, the last having antimicrobial properties [19]. The peroxiredoxin and glutathione S transferase are two antioxidant enzymes involved in the elimination of free radicals, potentially dangerous to the cells of all living organisms [20,21,22]. The hypopharyngeal glands are the main producers of RJ and glucose oxidase, glutathione S transferase S1, and peroxiredoxin 2540 have been identified in their proteome [23,24,25,26,27,28]. Glucose oxidase and glutathione S-transferase are also present in nectar [29,30,31], while peroxiredoxin has been detected in pollen [32]. It is therefore difficult to know whether these enzymes are produced by the bee and secreted in RJ or whether they are of vegetal origin. The post-production progressive decrease in abundance of these proteins could be due to the presence of proteases. In RJ activities of a carboxypeptidase A-like and a chymotrypsin-like, proteins were identified [33]. The presence of proteases in RJ has been previously hypothesized and discussed by several authors [8,10,34] but only a trypsin-like protease [8] and two serine-proteinase in RJ derived from thoracic glands [2] were identified.

Although RJ has been studied for a long time and arouses interest in quantitative aspects, their enzymatic activity through its shelf life, is still a neglected topic [35,36,37]. However, the presence of glucose oxidase and antioxidant enzymes activities can affect antimicrobial and antioxidant properties of RJ [19,21]. At the same time, the increase in proteases activity could determine proteins degradation resulting in a loss of product quality [10]. For these reasons, the measure of the enzymatic activity in RJ may be a useful tool for quality product assessment during its shelf life. The aim of this preliminary work was to investigate the enzymatic activity of glucose oxidase, five proteases (chymotrypsin, trypsin, N-aminopeptidase, carboxypeptidase A and B) and two antioxidant enzymes (peroxiredoxin and glutathione S transferase) in RJ under different storage condition (i.e., refrigeration at 4 °C and deep-freezing at −18°) and times.

## 2. Materials and Methods

### 2.1. Samples

The investigation was carried out on 5 samples of RJ for year produced in 2017, 2018, 2019, 2020 and on 6 samples produced in 2021. Samples collected in 2021 represented the control of each enzymatic investigation. For each year, samples were collected from the same beekeepers, produced in the same geographical areas and analyzed at the same time, belonging to the same commercial batches. The 2017, 2018 and 2019 samples were stored at 4 °C from production to analysis (Table 1). Samples collected in 2020 and 2021 were split at production time to store the same sample both at 4 °C (refrigeration, R) and at −18 °C (deep-freezing, F) until lab processing. In the text, ‘treatment’ is referred to as the type of storage (temperature and time of storage).

### 2.2. Protein Extraction and Quantification for Enzymatic Investigations

A total of 100 mg of sample were incubated with 150 µL of 0.1 M phosphate buffer pH7.2 with 1% Triton X-100. After centrifugation at 2500 g for 10 min, the supernatant was collected. To the resulting pellet, 150 µL of 0.1 M phosphate buffer pH7.2 was added and centrifuged again at 2500 g for 10 min. The supernatant was added to that taken from the first centrifuge and stored at −20 °C until analysis. The protein concentration of each sample was determined by Bradford’s reagent [38], using γ-globulin to create the standard curve.

### 2.3. Measurement of Glucose Oxidase Activity

The glucose-oxidase activity was determined according to Cohen [39] and Sagona et al. [40] with some modifications. For each protein extract, 1.5 mL of solution containing 100 mM Hepes buffer pH7, 0.1 mM EDTA, 5 mM D-glucose, Diaminobenzidine (DAB) (0.18 mg/mL), and horseradish peroxidase (HRP) (0.02 mg/mL) was prepared. Absorbance was obtained at a wavelength of λ = 352 nm at times 0 and 120 min. All enzymatic analyses were conducted using a Lambda 25 UV/VIS spectrophotometer (Perkin Elmer, Milano, Italy).

### 2.4. Measurement of Proteases’ Activity

Chymotrypsin-like activity was determined in accordance with Hummel protocol [41] and Wirnt [42], using 1.18 mM N-Benzoyl-L-Tyrosine Ethyl Ester (BTEE) as substrate. To a solution containing 0.52 mL of 80 mM Tris HCl buffer with 100 mM calcium chloride pH7.8, and 0.48 mL of BTEE, 33 µL of sample was added. Absorbance was recorded at a wavelength of 256 nm for 3 min.

Trypsin-like activity was determined using 20 mM N-α-Benzoyl-DL-arginine-p-nitroanilide (BAPNA) as substrate, according to Bisswanger [43] with some modifications. To a solution containing 0.7 mL distilled water, 0.18 mL BAPNA and 0.10 mL 0.3 M potassium phosphate buffer pH 8.0, 20 µL of protein extract was added. Absorbance was recorded at a wavelength of 405 nm.

N-aminopeptidase-like activity was determined using L-alanine-p-nitroanilide as a substrate according to the protocol of Coppola et al. [44]. A volume of 10 µL of sample was added to 1 mL of a test solution consisting of 2.04 mM L-alanine-p-nitroanilide in 0.1 of phosphate buffer pH7.2 and incubated at 37 °C for 10 min. The reaction was stopped with 3 mL of cold glacial acetic acid. The absorbance was measured at 384 nm. Activity was determined using the molar extinction coefficient of p-nitroanilide (ε = 8.8 M^−1^ cm^−1^).

Carboxypeptidase-like A and B activities were determined according to the protocol of Grogan and Hunt [45]. For carboxypeptidase A activity, 2.9 mL of a solution containing 1 mM Hippuryl-L-phenylalanine (HPA) dissolved in 25 mM Tris HCl buffer pH7.5 and containing 500 mM NaCl was added to 0.1 mL of sample protein extract. For carboxypeptidase B, 2.9 mL of a solution containing 1 mM Hippuryl-L-arginine (HA) dissolved in 25 mM Tris HCl buffer pH 7.65 and containing 100 mM NaCl was added to 0.1 mL of sample protein extract. The change in absorbance at 254 nm as a function of time was recorded as an index of activity for both enzymes.

### 2.5. Measurement of Antioxidant Enzymes’ Activity

The peroxiredoxin activity was determined according to the protocol of Ali and Hadwan [46]. Hydrogen peroxide 2.1 mM was added to a solution containing 50 mM phosphate buffer (PB) pH7, 10 mM sodium azide, 2.1 mM 1,4-dithio-DL-threitol (DTT) and the sample protein extract and incubated at 37 °C for 3 min. FOX reagent consisting of two reagents A and B, in a ratio 1:100, was added to an aliquot of this mixture. FOX reagent A contains 25 mM ammonium iron sulphate in 25 mM sulfuric acid, while FOX reagent B consists of 100 mM sorbitol and 125 µM xylenol orange in distilled water. The solutions were incubated at room temperature for 30 min. The change in absorbance was measured at 560 nm. The glutathione S-transferase activity was determined according to Habig et al. [47] with some modifications. A solution consisting of 1.2 mL 0.1 M phosphate buffer pH 6.5, 50 µL 1 mM 1-chloro-2,4-dinitrobenzene (CDNB) in methanol and 200 µL distilled water was added to the protein extracts. A further 100 µL 5 mM reduced glutathione (GSH) was added to the cuvettes prior to spectrophotometric reading. The absorbance of individual samples was recorded at a wavelength of 340 nm for 5 min.

### 2.6. Statistical Analysis

Data were processed using JMP 7 software (Cary, NC: SAS Institute, 2008). After checking the distribution of data using the Shapiro-Wilk test, normally distributed data were analyzed using ANOVA test followed by Tukey’s test was applied or t-student test. Not-normally distributed data were analyzed by the non-parametric Kruskal-Wallis H-test, followed by the post hoc Wilcoxon test. Differences in glucose oxidase, carboxypeptidase-like A, trypsin-like, N-aminopeptidase-like, carboxypeptidase-like B activities in RJ samples stored for different period at 4 °C, were analyzed using the non-parametric Kruskal-Wallis H-test, followed by the post hoc Wilcoxon test while for Chymotrypsin-like, peroxiredoxin, and glutathione S-transferase ANOVA test followed by Tukey’s test were used. Differences in glucose oxidase, carboxypeptidase-like A, trypsin-like, N-aminopeptidase-like, carboxypeptidase-like B activities between RJ samples refrigerated and frozen for two months and 1 year were analyzed using the non-parametric Wilcoxon test, while for chymotrypsin-like, peroxiredoxin, and glutathione S-transferase activities, the T-Student test was used. *p*-values lower than 0.05 were considered significant.

## 3. Results

Glucose oxidase activity in refrigerated RJ samples resulted significantly higher after two months of storage than from one to four years (*p* = 0.015). No differences among RJ samples stored from one to four years at 4 °C were recorded. Significantly higher carboxypeptidase-like A activity was recorded at two-months and two years old samples with respect of the other investigated time points (*p* = 0.007; Table 2). Chymotrypsin-like, trypsin-like, N-aminopeptidase-like, carboxypeptidase-like B, peroxiredoxin and glutathione S transferase activities did not show statistical difference in refrigerated samples in relation to the storage time.

The analysis of 2020 and 2021 samples enabled a direct comparison of different storage treatments on the same batch of production: refrigeration at 4 °C determined a higher glucose oxidase activity in RJ samples after two months of storing rather than after 1 year (*p* = 0.009), while no differences were detected among samples stored at −18 °C for different period of time. Furthermore, after 1 year of refrigerated storage, RJ samples showed significant lower glucose oxidase activity than those deeply frozen (*p* = 0.047; Table 3).

RJ samples stored at 4 °C for 1 year also showed lower carboxypeptidase-like A activity than those stored for 2 months (*p* = 0.006). No differences in carboxypeptidase-like A activity were recorded among samples stored at −18 °C for different period of time (1 year vs. 2 months) (*p* > 0.05). After 2 months of storage carboxypeptidase A-like activity was higher in samples stored at 4°C than in those stored at −18 °C (*p* = 0.025), while after 1 year result higher in samples stored at −18 °C than those at 4 °C (*p* = 0.047).

Trypsin-like activity resulted higher in RJ samples stored at 4 °C for 1 year than in those stored for 2 months (*p* = 0.018), while no differences were recorded among samples stored at −18 °C for different period of time (1 year vs. 2 months) as well as for the same period of time under different storage conditions. No differences were also recorded on chymotrypsin-like, N-aminopeptidase-like, carboxypeptidase-like B, peroxiredoxin and glutathione S transferase activities among samples stored for different period of time (1 year vs. 2 months) and for the same period of time under different storage conditions.

## 4. Discussion

To the best of our knowledge, the present study investigates for the first time the potential activity of glucose oxidase, five proteases (chymotrypsin, trypsin, N-aminopeptidase, carboxypeptidase A and B) and two antioxidant enzymes (peroxiredoxin and glutathione S transferase) in RJ stored under different condition (+4 °C and −18 °C) for different period of time.

Our results indicate that refrigeration at 4 °C determined a significant decrease in glucose oxidase activity after one year of storage to then remains stable for the subsequently years of storage under similar conditions. A high activity of glucose oxidase may be a detrimental factor for the quality of RJ, since it can lead to a degradation of the lipids which represent the 3–8% of its composition [4]. It is known that hydrogen peroxide, generated from the glucose oxidase activity, causes lipid peroxidation [48]. On the other hand, hydrogen peroxide has an antimicrobial action which could prevent spoilage of the product while stored at 4 °C [49,50]. However, the lower activity of glucose oxidase recorded could be due to partial inhibition, or fewer active molecules because the others are degraded; thus, further investigation to assess the period over which glucose oxidase activity is preserved is desirable. The intermediate product of glucose oxidase, glucono-lactone is a weak competitive inhibitor of glucose oxidase, and the activity of the enzyme also declines in the presence of aldohexoses such as D-arabinose and 2-deoxy-D-glucose which act as competitive inhibitors [51]. The possible presence of these competitive inhibitor could contribute to the decrease in glucose oxidase activity. After one year of storage, decrease in glucose oxidase activity also occurred in frozen RJ, keeping though high level of activity than at 4 °C. Therefore, the decrease in glucose oxidase activity after one year of storage at 4 °C suggests that this enzyme may be a good indicator of RJ freshness, while also highlighting the capacity of frozen storage to maintain high level of glucose oxidase activity. Our results suggest focusing further investigations within the first year from production by narrowing the observation intervals.

The assessment of chymotrypsin, trypsin, N-aminopeptidase, carboxypeptidase A and B activity values obtained in this study to investigate possible protein degradation allow us to detect and quantify for the first time the activity of these proteases in Royal Jelly. To the best of our knowledge, these proteases have not yet been identified in the RJ proteome [2,7,8,9,10,11,12,13,34]. Fujita et al. [2] identified two serine-proteinase in RJ derived by thoracic glands and only trypsin-like protease was previously found in RJ by Kamakura et al. [8]. In honeybees, chymotrypsin-1 with molecular mass 28 KDa, trypsin-1 29 Kda, N-aminopeptidase 90–130 KDa, carboxypeptidase A 38 KDa, carboxypeptidase B-like 48 KDa [33,52,53] have been identified. No protease activity was detected in nectar using casein as substrate [54]. Conversely, trypsin-like, chymotrypsin-like, carboxypeptidase A and B-like, and aminopeptidase activities, were found in pollen extract [45,55]. Therefore, the protease activity detected in RJ in this study and not by proteomic approach leads us to develop two hypotheses: (I) proteases are a component of RJ secreted by the honeybee as an expression of its genome but were not previously detected because of applied technical constraints. This hypothesis could account for the absence of low molecular weight proteases which may be lost in molecular sieves, such as the SDS page with large meshes [56]. (II) Proteases are not an expression of the honeybee genome but rather present as contaminants by the honeybee gut content (i.e., pollen, nectar). Pollen is a natural pollutant of RJ and its low presence in samples may not allow for the detection of proteases where MRJPs are predominant, but under optimal conditions they might perform their activity. This could have implications that go beyond assessing the product freshness, for instance helping to detect food frauds, as an artificially produced RJ would not detect proteases. Among RJ proteases investigated in this study, only carboxypeptidase A-like showed significant variation in activity level in relation to the storage period length at 4 °C with higher activity level recorded after two months. The high variability of carboxypeptidase A-like activity could be due to the difference in RJ samples production (i.e., collection time) but we cannot exclude that geographical or botanical features might play a role as well. The method of collection and analysis can be furtherly improved, but the fact remains that it could be a freshness and authenticity indicator for RJ. Although carboxypeptidase A is an exopeptidase, that is specific for catalyzing the cleavage of certain carboxyl-terminal peptide bonds in peptides and proteins; therefore, it may contribute to the degradation of glucose oxidase. However, as its activity increases with increasing glucose oxidase activity, it would appear not to be involved in its degradation [57]. This result indicates that also carboxypeptidase A could be a good freshness indicator for RJ. However, further investigation on carboxypeptidase A-like activity over the time on RJ samples of same geographical or botanical features produced using the same method are desirable in order to better understand the activity trend of these proteases.

Concerning the products of glucose oxidase, hydrogen peroxide and their ability to degrade lipids, two antioxidant enzymes known to be present in RJ were investigated. Peroxiredoxin enzymes are antioxidant enzymes belonging to a family of cysteine-dependent peroxidases that react with hydrogen peroxide, aliphatic and aromatic hydroperoxide substrates, and peroxynitrite [16], while glutathione S transferase enzymes have function of detoxification of electrophiles by glutathione conjugation [58]. Both of these enzymes do not change their activity during refrigerated storage; this could be an advantage as they represent an always active system for the elimination of free radicals. However, they are not an indicator of the freshness of RJ, within the examined timeframe.

According to time and storage condition, enzymatic activity in RJ samples showed a high variability, probably due to different collection time as well as geographical or botanical features. Despite this high variability, the storage of RJ at 4 °C for different period determined significant differences in enzymatic activity compared to treatment at −18 °C. After 1 year of storage at 4 °C the glucose oxidase activity in RJ undergoes to a significant decrease (77%), while −18 °C storage preserves it to a greater extent. Again, two hypotheses could be suggested to explain this difference: (i) the freezing treatment might have removed the action of a potential inhibitor of the glucose oxidase by modification of inhibitor structure; (ii) at 4 °C proteases degrade glucose oxidase, thus reducing its activity. This second hypothesis is probably the most reliable, but further investigation is desirable.

Moreover, the deep-freezing treatment determined an early low reduction in carboxypeptidase A-like activity in RJ after two months of storage compared to refrigeration and remain constant after 1 year. Differences in carboxypeptidase A-like activity trend in frozen rather than refrigerated RJ suggest enzymatic inhibition by the product, but further investigations are needed.

Trypsin-like activity increased significantly during refrigerated conservation; this suggests that trypsin may somehow be involved in the digestion of glucose oxidase as well as could be hypothesized that trypsin-like digests the glucose oxidase potential inhibitor. Further investigations are needed to investigate the activity of this important enzyme and its eventual link with others present in RJ.

Both antioxidant enzymes investigated (peroxiredoxin and glutathione S transferase) did not result in any change in their activity under both treatment conditions, thus confirming that they cannot be considered as possible indicators of freshness in RJ.

## 5. Conclusions

In conclusion, results obtained indicate that glucose oxidase and carboxypeptidase A-like may be good indicators of freshness in Royal Jelly, especially within the first year from production, since they returned a consistent trend in their activity, regardless the storage treatment applied. Results obtained also indicate that deep-freezing could be a better alternative storage method to refrigeration since it determines a high preservation of glucose oxidase and carboxypeptidase A-like after 1 year of storage. Investigation on the enzymatic activity trend for longer period in RJ stored ad −18° are desirable. Furthermore, evidence recorded in this study might indicate that for management and commercial purposes, the cold chain at −18 °C could be repeatedly interrupted with short exposures at 4 °C without negatively impacting on overall product quality and biological functions, but the effect of repeated thawing on RJ composition still has to be assessed.

## Figures and Tables

**Table 1 foods-11-03165-t001:** Relative amount of RJ samples investigated per treatment (4 °C, refrigeration “R”; −18 °C, deep-freezing “F”); each of two months and one-year-old sample was analyzed both for R and F.

Collection Year	Storage Time	R	F
2017	4 years	5	
2018	3 years	5	
2019	2 years	5	
2020	1 year	5	5
2021 (control)	2 months	6	6
**Total**		26	11

**Table 2 foods-11-03165-t002:** Statistical differences of enzymatic activity in RJ samples stored for different period (2 months = 2m; 1 year = 1y; 2 years = 2y, 3 years = 3y and 4 years = 4y) at 4 °C. Values are expressed as average ± SE (median). Significant differences are indicated with superscript different letters and marked in bold.

	RJ Storage Time					
Enzyme	2m	1y	2y	3y	4y	*p-Value*
**Glucose oxidase** (U/mg proteins)	**274 ± 70 ^a^** **(279)**	**62 ± 21 ^b^** **(45)**	**91 ± 6 ^b^** **(91)**	**129 ± 24 ^b^** **(130)**	**93 ± 30 ^b^** **(98)**	*0.015*
**Carboxypeptidase-like A** (U/mg proteins)	**33.7 ± 10.5 ^a^** **(22.8)**	**3.9 ± 1.3 ^c^** **(5.2)**	**13.3 ± 2.3 ^b^** **(13.1)**	**4.5 ± 1.4 ^c^** **(4.3)**	**6.2 ± 3.2 ^bc^** **(1.6)**	*0.007*
**Trypsin-like** (U/mg proteins)	0.40 ± 0.02 (0.40)	0.50 ± 0.03 (0.48)	0.45 ± 0.03 (0.41)	0.47 ± 0.02 (0.47)	0.49 ± 0.05 (0.49)	*0.136*
**Chymotrypsin-like** (U/mg proteins)	0.07 ± 0.02 (0.06)	0.03 ± 0.01 (0.01)	0.08 ± 0.02 (0.06)	0.09 ± 0.02 (0.10)	0.07 ± 0.02 (0.06)	*0.271*
**N-aminopeptidase-like** (U/mg proteins)	0.58 ± 0.19 (0.44)	0.35 ± 0.06 (0.34)	0.30 ± 0.03 (0.32)	0.33 ± 0.03 (0.30)	0.32 ± 0.04 (0.31)	*0.692*
**Carboxypeptidase-like B** (U/mg proteins)	0.29 ± 0.21 (0.04)	0.20 ± 0.12 (0.04)	0.37 ± 0.30 (0.02)	0.19 ± 0.15 (0.00)	0.77 ± 0.22 (0.59)	*0.206*
**Peroxiredoxin** (U/µg proteins)	4.44 ± 0.24 (4.43)	3.41 ± 0.91 (3.84)	5.00 ± 0.70 (5.22)	4.94 ± 0.68 (5.52)	4.32 ± 0.46 4.37	*0.413*
**Glutathione S transferase** (gst mM/min/µg proteins)	116 ± 19 (111)	120 ± 10 (109)	119 ± 16 (117)	87 ± 20 (84)	102 ± 12 (107)	*0.575*

**Table 3 foods-11-03165-t003:** Statistical differences in enzymatic activity in RJ samples refrigerated (R) at 4 °C and frozen (F) at −18 °C for two months (2m) and 1 year (1y). All values are expressed as average ± SE (median). Significant differences are marked in bold.

Enzyme	2mR *vs.* 1yR	2mF *vs.* 1yF	2mR *vs.* 2mF	1yR *vs.* 1yF
**Glucose oxidase** **(U/mg proteins)**	274 ± 70(279) *vs.* 62 ± 21(45) ***p = 0.009;***	772 ± 436(272) *vs.* 433 ± 140(392) *p = 0.754;*	274 ± 70(279) *vs.* 772 ± 436(272) *p = 0.754;*	62 ± 21(45) *vs.* 433 ± 140(392) ***p = 0.047;***
**Carboxypeptidase A-like** **(U/mg proteins)**	33.7 ± 10.5(22.8) *vs.* 3.9 ± 1.3(5.2) ***p = 0.006;***	9.4 ± 5.1(4.0) *vs.* 9.7 ± 3.4(7.5) *p = 0.584;*	33.7 ± 10.5(22.8) *vs.* 9.4 ± 5.1(4.0) ***p = 0.025;***	3.9 ± 1.3(5.2) *vs.* 9.7 ± 3.4(7.5) ***p = 0.047;***
**Trypsin-like** **(U/ mg proteins)**	0.40 ± 0.02(0.40) *vs.* 0.50 ± 0.03(0.48) ***p = 0.018;***	0.49 ± 0.04(0.46) *vs.* 0.50 ± 0.03(0.48) *p = 0.718;*	0.40 ± 0.02(0.40) *vs.* 0.49 ± 0.04(0.46) *p = 0.109;*	0.50 ± 0.03(0.48) *vs.* 0.50 ± 0.03(0.48) *p = 0.917;*
**Chymotrypsin-like** **(U/mg proteins)**	0.07 ± 0.02(0.06) *vs.* 0.03 ± 0.01(0.01) *p = 0.120;*	0.11 ± 0.02(0.11) *vs.* 0.05 ± 0.02(0.06) *p = 0.133;*	0.07 ± 0.02(0.06) *vs.* 0.11 ± 0.02(0.11) *p = 244;*	0.03 ± 0.01(0.01) *vs.* 0.05 ± 0.02(0.06) *p = 0.348;*
**N-aminopeptidase-like** **(U/mg proteins)**	0.58 ± 0.19(0.44) *vs.* 0.35 ± 0.06(0.34) *p = 0.361;*	0.43 ± 0.06(0.47) *vs.* 0.38 ± 0.08(0.28) *p = 0.465;*	0.58 ± 0.19(0.44) *vs.* 0.43 ± 0.06(0.47) *p = 0.749;*	0.35 ± 0.06(0.34) *vs.* 0.38 ± 0.08(0.28) *p = 0.917;*
**Carboxypeptidase B-like** **(U/mg proteins)**	0.29 ± 0.21(0.04) *vs.* 0.20 ± 0.12(0.04) *p = 0.848;*	0.13 ± 0.13(0.00) *vs.* 0.69 ± 0.65(0.00) *p = 0.416;*	0.29 ± 0.21(0.04) *vs.* 0.13 ± 0.13(0.00) *p = 0.295;*	0.20 ± 0.12(0.04) *vs.* 0.69 ± 0.65(0.00) *p = 0.738;*
**Peroxiredoxin** **(U/µg proteins)**	4.44 ± 0.24(4.43) *vs.* 3.41 ± 0.91(3.84) *p = 0.264;*	3.89 ± 0.82(3.84) *vs.* 3.41 ± 0.51(3.65) *p = 0.645;*	4.44 ± 0.24(4.43) *vs.* 3.89 ± 0.82(3.84) *p = 0.535;*	3.41 ± 0.91(3.84) *vs.* 3.41 ± 0.51(3.65) *p = 0.996;*
**Glutathione S transferase** **(gst mM/min/µg proteins)**	116 ± 19(111) *vs.* 120 ± 10(109) *p = 0.868;*	132 ± 19(130) *vs.* 77 ± 21(56) *p = 0.081;*	116 ± 19(111) *vs.* 132 ± 19(130) *p = 0.540;*	120 ± 10(109) *vs.* 77 ± 21(56) *p = 0.108*

## Data Availability

The data used to support the findings of this study can be made available by the corresponding author upon request.

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
