# Peer review of "Impact of Different Storage Temperature on the Enzymatic Activity of Apis mellifera Royal Jelly"

_foods, 2022, doi:10.3390/foods11203165_

Round 1

Reviewer 1 Report

The manuscript describes the results of analyzes aimed at determining the activities of 5 proteases and two antioxidant enzymes in royal jelly, subjected to different preservation methods.

The article is clear and well describes materials, methods and results. The introduction is satisfactory and interests the reader. In any case, some small changes should be made to improve the description of the results 

Line 169:  table should only contain a short title; all the other indication (lines 171-177) can be inserted in the caption below and/or in the text above.

The same goes for Table 3.

Again as regards the data in table 3, usually the values of χ² and df are not entered, but simply the value of P, and therefore the statistical significance.

Author Response

Response to Reviewer #1 comments

Authors wish to thank the reviewer for its suggestions. All the suggested changes have been incorporated in the manuscript and detailed point-by-point response to each specific comment has been given below. All changes are tracked in the manuscript.

The manuscript describes the results of analyzes aimed at determining the activities of 5 proteases and two antioxidant enzymes in royal jelly, subjected to different preservation methods. The article is clear and well describes materials, methods and results. The introduction is satisfactory and interests the reader. In any case, some small changes should be made to improve the description of the results

Line 169:  table should only contain a short title; all the other indication (lines 171-177) can be inserted in the caption below and/or in the text above.

Authors agree with the reviewer and following the suggestion provided indication from line 171 to 177 have been removed from table caption and included in the statistical analysis section

The same goes for Table 3.

Following the reviewer suggestion caption of Table 3 has been simplified and indication on statistical analysis performed included in the statistical analysis section

Again as regards the data in table 3, usually the values of χ² and df are not entered, but simply the value of P, and therefore the statistical significance.

Following the reviewer suggestion values of χ² and df have been removed from both tables 2 and 3.

Reviewer 2 Report

Dear authors,  

The topic of finding the effect of storage conditions and time, such as royal jelly, is exciting and relevant in the light of bee product development. 

Introduction

Point 1: Please include the reason for the choice of marker for RJ quality. Why the authors focused only on enzymatic activity?

Materials and Methods 

Point 2: Why did the author use the royal jelly from a different production year to compare the enzymatic activity? The royal jelly composition and properties change with the production year. The proper experiment could use the same batch to approximate the effect of storage time and temperature.

Point 3: Apparently, the royal jelly products in the market indicated a shelf-life of about six months in the refrigerator to three years in a deep freezer. Why did the authors use an age-old one, like four years of storage RJ? Did the authors consider the freeze-dried RJ? 

Point 4: Why did the author focus only on enzymatic activity? How about the quantity of the 10-HDA and/or antimicrobial and antioxidant activity? Obviously, the enzyme in the royal jelly slightly reflexes the quality of the royal jelly. Yet, the physicochemical characterization of RJ should be conducted.

Author Response

Response to Reviewer #2 comments

Authors wish to thank the reviewer for its insightful suggestions which significantly improved the

manuscript. All the suggested changes have been incorporated in the manuscript and detailed point-

by-point response to each specific comment has been given below. All changes are tracked in the

manuscript.

The topic of finding the effect of storage conditions and time, such as royal jelly, is exciting and relevant in the light of bee product development.

Introduction

Point 1: Please include the reason for the choice of marker for RJ quality. Why the authors focused only on enzymatic activity?

Authors thank the reviewer for this insightful suggestion. Following the reviewer observation introduction section has been re-formulated to better explain the reason for the choice of marker for RJ quality.

Materials and Methods

Point 2: Why did the author use the royal jelly from a different production year to compare the enzymatic activity? The royal jelly composition and properties change with the production year. The proper experiment could use the same batch to approximate the effect of storage time and temperature.

In this study beekeepers and areas of RJ production were chosen to be always the same throughout the 5 years of investigation. The different years of production were chosen for two main reasons: I) compare all years of production each other to outline the effect of time of storage randomly; II) maximize the opportunity to investigate enzymatic activity in RJ kept in the same storage condition for 4 years with the fresh one. In this way the application of statistical analysis resulted to be able to discriminate the enzymatic activity of the fresh RJ from the other different years of storage, indicating that the enzymatic activity was not different in samples stored for one year compared to those stored for 4 years.

Point 3: Apparently, the royal jelly products in the market indicated a shelf-life of about six months in the refrigerator to three years in a deep freezer. Why did the authors use an age-old one, like four years of storage RJ? Did the authors consider the freeze-dried RJ?

Authors are aware on the indication about RJ shelf-life. However, RJ shelf-life were determined based on humidity, 10-HAD and furosine (Bogdanov, S. 2011. The royal jelly book. Bee Product Science.) while in this study Authors wanted to investigate and describe the enzymatic activity in RJ stored in different conditions for different times.

Point 4: Why did the author focus only on enzymatic activity? How about the quantity of the 10-HDA and/or antimicrobial and antioxidant activity? Obviously, the enzyme in the royal jelly slightly reflexes the quality of the royal jelly. Yet, the physicochemical characterization of RJ should be conducted.

Authors focused only on enzymatic activity since quantity of the 10-HAD, antimicrobial and antioxidant activity have already been partially investigated (Antinelli et al., 2003; Bărnuţiu et al., 2011; Pavel et al., 2014; Alreshoodi & Sultanbawa 2015; Balkanska et al., 2017) while no information are available on enzymatic activity. Authors agree with the reviewer about the importance to perform physicochemical characterization of RJ and this could be the topic of a next investigation.